# Preparation of a Low-Protein-Fouling and High-Protein-Retention Membrane via Novel Pre-Hydrolysis Treatment of Polyacrylonitrile (PAN)

**DOI:** 10.3390/membranes13030310

**Published:** 2023-03-08

**Authors:** Dong Xu, Guangyao Pan, Yutong Ge, Xuan Yang

**Affiliations:** National University of Singapore (Suzhou) Research Institute, Suzhou 215123, China

**Keywords:** membrane preparation, polyacrylonitrile, hydrolysis treatment, anti-fouling modification, high protein retention

## Abstract

The attainment of high-protein-retention and low-protein-fouling membranes is crucial for industries that necessitate protein production or separation process. The present study aimed to develop a novel method for preparing polyacrylonitrile (PAN) membranes possessing a highly hydrophilic and negatively charged surface as well as interior structure. The method involved a pre-hydrolysis treatment during the preparation of the PAN dope solution, followed by phase inversion in an alkaline solution. Chemical and material characterization of the dopes and membranes uncovered that the cyclized PAN structure served as a reaction intermediate that facilitated strong hydrolysis effect during phase inversion and homogeneously formed carboxyl groups in the membrane’s interior structure. The resulting membrane showed a highly hydrophilic surface with a contact angle of 12.4° and demonstrated less than 21% flux decay and more than 95% flux recovery during multi-cycle filtration of bovine serum albumin (BSA) solution, with a high protein rejection rate of 96%. This study offers a facile and effective alternative for preparing PAN membranes with enhanced antifouling and protein-retention properties.

## 1. Introduction

The process of separating and purifying proteins is crucial in food, biomedical, and pharmaceutical industries. The separation of proteins has been realized through a variety of techniques, including precipitation, crystallization, chromatography, ultrafiltration, adsorption [1,2]. Membrane-based technologies have become one of the favored solutions for separating and purifying proteins due to their high separation efficiency, facile implementation, and cost-effectiveness [3,4]. However, the fouling of membranes by proteins presents a longstanding issue in various industries, as it may lead to a reduction in membrane permeability and escalation of operational cost, particularly in membrane processes that require high protein retention. The selectivity of proteins by membranes is regulated by both the size sieving and electrostatic exclusion effects [5,6]. The size sieving effect is mainly determined by the membrane’s pore size, while the electrostatic exclusion effect is primarily controlled by the membrane’s charge. In recent years, whey protein has gained popularity in the health food market and is commonly consumed as a protein supplement. The main proteins found in whey, such as β-lactoglobulin, α-lactalbumin, BSA, have an isoelectric point within the range of weak acidity [7], which results in negatively charged protein colloids when the membrane filtration is carried out in a neutral environment. Therefore, a negatively charged membrane surface is considered to have a higher capability to resist fouling, and to retain proteins that have the same sign of charge, due to the electrostatic repulsion [8].

Several studies have documented the synthesis of negatively charged membranes that exhibit both high retention and improved anti-fouling properties for the filtration of proteins. Kumar and Ulbricht blended sulfonic acid functionalized multi-walled carbon nanotubes into partially sulfonated poly(arylene ether sulfone) to produce a negatively charged membrane surface within the pH range of 3–10, with high hydrophilicity and low protein adsorption propensity. The prepared membrane also showed a high degree of selectivity between BSA and myoglobin [9]. Arunkumar and Etzel carried out modifications on commercial cellulose membranes by introducing negatively charged sulfonic acid groups onto the membrane surface. The modification facilitated the use of wide-pore membranes with high flux without compromising protein retention. Their results showed that the negatively charged 100 kDa membrane had the same protein retention rate as the 10 kDa unmodified membrane, but with a flux that was at least two times higher [10]. In a recent study, Ye et al. investigated the modification of negatively charged poly(acrylic acid) onto a ceramic membrane, and their findings presented enhancements in the membrane’s protein-resistant and easy-cleaning characteristics. Specifically, the modified membrane showed a 60% increase in stable normalized flux and a 55% increase in flux recovery ratio [11]. These studies have demonstrated enhancements in protein retention and fouling resistance; however, there is a scarcity of reporting on interior or homogeneous anti-fouling modifications. For instance, during the fractionation of whey protein, the selective sieving of protein must enable small non-target proteins to penetrate the membrane separation layer and may lead to irreversible internal pore fouling.

PAN is a cost-effective material possessing desirable physiochemical characteristics (such as good thermal stability, high modulus, high resistance to chemicals, especially to acids) and has been extensively utilized in the manufacture of membranes [12]. The utilization of PAN-based membranes for protein separation has a long history. The presence of rich polar nitrile groups in PAN molecules not only endows the PAN membrane with inherent hydrophilicity, but also provides ideal sites for further modifications. The alkaline hydrolysis of PAN is one of the simplest modification methods to effectively convert nitrile groups into hydrophilic and negatively charged carboxylate groups [13]. From the research of the PAN hydrolysis mechanism, Karpacheva et al. identified that the intermediate hydrolysis product of cyclized PAN in short sequences, as shown in Figure 1, exhibited greater reactivity compared to the nitrile groups [14]. Over the years, research on the hydrolyzed PAN intermediate has been predominantly centered around elucidating the underlying principles, with a paucity of attention paid to its practical applications. Moreover, previous investigation has mainly utilized post-hydrolysis treatment, which limits the modification to the membrane surface.

In this study, different from the frequently used post-hydrolysis treatment of PAN membranes, we introduce a new pre-hydrolysis method for preparing highly hydrophilic and negatively charged PAN-based membranes with homogeneous modification effect on both the surface and interior structure of the membrane. The simple modification method endowed the prepared membrane with exceptional resistance to protein fouling and high protein retention. To conduct the modification, PAN polymer was at first pre-hydrolyzed during its dope preparation to partially produce cyclized PAN intermediate. The pre-hydrolyzed PAN dope was then used to fabricate PAN-based membranes through the alkali-induced phase inversion process as we previously described [15]. The pre-hydrolyzed PAN dope was subjected to FTIR and NMR characterization. The morphology of the resulting membranes was analyzed using SEM, their surface functional group was evaluated through FTIR spectrometer, and their wetting property was assessed using a contact angle analyzer. The protein retention and anti-fouling performances of the prepared membranes were investigated by the filtration of protein solution using BSA as the model retentate. 

## 2. Materials and Methods

### 2.1. Materials

PAN with an average molecular weight of approximately 150,000 from Sinopharm was dried under vacuum at room temperature for 24 h before use. Methanol, isopropanol and sodium hydroxide (NaOH) were purchased from Adams (Shanghai, China). Dimethyl sulfoxide (DMSO) and phosphate buffered saline (PBS) solution were purchased from Macklin (Shanghai, China). BSA fraction V was purchased from BioFroxx (Guangzhou, China). All reagents were used as received without further purification. Deionized (DI) water with a conductivity of 18.2 MΩ cm^−1^ was obtained from a water purification system (5AS, Biosafer, Nanjing, China).

### 2.2. Membrane Preparation

The general membrane preparation process is illustrated in Figure 2. To prepare unmodified PAN membrane dope, the dry PAN polymer was dissolved in DMSO at a concentration of 14% by weight. For the preparation of the pre-hydrolyzed PAN dope, referred to as hPAN dope, the PAN polymer was completely dissolved in DMSO first. Afterwards, a designed quantity of an alkali solution, made by dissolving NaOH in methanol to have a molar concentration of hydroxyl ions to be 5 N, was added to the dope. The pre-hydrolysis reaction was then carried out under the protection of argon, lasting for 6 h at 50 °C. Based on the molar concentration of the added hydroxide to the total nitrile groups, the hPAN dope solutions were referred to as hPAN3000, hPAN5000 and hPAN7000.

Before casting membranes, the prepared dope solutions were subjected to centrifugation at 10,000 rpm for 5 min to eliminate any trapped air bubbles. The membranes were cast on a glass plate using a film applicator. The membrane thickness was set at 300 μm. The unmodified PAN membrane was precipitated in DI water for 48 h. Initially, the pre-hydrolyzed PAN-based membranes were precipitated in a coagulation bath containing 10% (*w/v*) NaOH solution for 4 h. Then, they were transferred to a DI water bath for another 48 h prior to testing. The temperature at which all the membranes precipitated was fixed at 25 °C. Based on the different dope solutions and coagulation baths used, the prepared membranes were labeled as presented in Table 1. The membranes used for the filtration test were kept in DI water, while those intended for characterization analyses were dried in a desiccator for at least 24 h before use.

### 2.3. Characterization of hPAN

The hPAN samples, including hPAN3000, hPAN5000, and hPAN7000, were prepared by dropwise precipitation in isopropanol. The obtained sample beads were collected through vacuum filtration and then dried in a fume hood before being stored in a desiccator. The functional groups of the hPAN samples were analyzed using a FTIR spectrometer (Nicolet iS5, Thermo Fisher Scientific, Waltham, MA, USA). The 1H NMR spectra of the unmodified PAN and hPAN7000 in DMSO-d6 were obtained using an NMR spectrometer (Avance III HD 500, Bruker, Billerica, MA, USA).

### 2.4. Characterization of Membranes

The morphology of the prepared membranes was characterized by an SEM (Quanta 250, Thermo Fisher Scientific, USA). Prior to the characterization, membrane samples were coated with gold nanoparticles by a sputter coater (108 auto, Cressington, Watford, UK) by applying a current of 30 mA for 45 s. The surface pore size of the membrane samples was determined by the open-source imaging processing proGram- ImageJ (ver. 1.53k, MD, USA). Prior to the cross-sectional observation, the membrane samples were subjected to freeze-fracturing in liquid nitrogen. The surface and interior functional groups of the prepared membranes were analyzed through FTIR spectra, using the same procedure as the hPAN sample characterization. The hydrophilicity of the prepared membranes was assessed through the measurement of water contact angle (CA) with a contact angle analyzer (FCA2000A, AFES, Shanghai, China). The captive bubble method based on Baek’s work [16] was employed to obtain stable CA values.

### 2.5. Filtration Experiment

The BSA solution with a concentration of 1000 mg∙L^−1^, was prepared for the protein filtration experiment. This was achieved by dissolving 1 g of BSA in 1 L of 0.01 M PBS solution (pH7.4). The concentration of BSA in both the membrane feed (*C_f_*) and permeate (*C_p_*) was determined using a UV-vis spectrophotometer (UV-3600, Shimadzu, Kyoto, Japan), by measuring the UV absorbance at a wavelength of 280 nm [17]. The BSA rejection rate (*R*) was calculated based on the following equation:(1)$$R\left(\%\right)=\left(1-\frac{C_{p}}{C_{f}}\right)\times 100\%$$

As illustrated in Figure 3, a dead-end filtration system, consisting of two separate paths of forward filtration and backwash, was utilized to conduct filtration tests on the prepared membranes. The effective filtration area of the membranes was 14.5 cm^2^. Prior to testing, each membrane was subjected to compaction with DI water for 2 h at a pressure of 0.12 MPa. Afterwards, a pure (DI) water flux (PWF, *J_w_*) test was performed for 30 min at a pressure of 0.1 MPa and a temperature of 22 ± 1 °C. The calculation of *J_w_* was performed based on the following equation:(2)$$J_{w}=\frac{Q}{\Delta t\times A}$$
where *J_w_* (L∙m^−2^∙h^−1^) represents the PWF, *Q* (L) is the volume of water that has permeated in a period of Δ*t* (h), *A* (m^2^) is the membrane area.

The BSA filtration experiment was conducted using the same apparatus and under the same applied pressure and ambient temperature as for the PWF measurement. Each filtration cycle consisted of three stages, namely pure water filtration, filtration with BSA solution, and backwash with pure water. The first run started with 30 min pure water filtration, during which the steady-state PWF was recorded as *J_w_*. Then, the feed water was switched to the BSA solution for 45 min and the final permeation flux was recorded as *J_p_*. Afterwards, the membrane was backwashed with pure water for 10 min. In order to carry out this process, the filtration cell was emptied, and the membrane was flipped upside down inside the membrane cell. This allowed the backwash flow to pass through the membrane in the direction from its permeate side to the feed side. In the second and third runs, the PWF (*J_r_*) measurement time was shortened to 10 min. The relative flux decline (*RFD*) was calculated based on the following equation: (3)$$RFD=\left(1-\frac{J_{p}}{J_{w}}\right)\times 100\%$$

The relative flux recovery (*RFR*) of PWF was calculated based on the following equation:(4)$$RFR=\frac{J_{r\;}}{J_{w}}\times 100\%$$
where *J_r_* represents the measured PWF after the 1st, 2nd and 3rd backwash stage.

The degree of fouling of the prepared membranes was evaluated in terms of the membrane resistance [18]. The total resistance was calculated based on the following equation:(5)$$R_{t}=R_{m}+R_{f}=R_{m}+R_{r}+R_{ir}$$
where *R_t_* (m^−1^) is the total resistance (m^−1^) of membrane, *R_m_* (m^−1^) is the intrinsic resistance, *R_f_* (m^−1^) is the total resistance of fouling, *R_r_* (m^−1^) is the reversible resistance of fouling, *R_ir_* (m^−1^) is the irreversible resistance of fouling. These resistance values can be calculated based on the following Equations (6)–(9):(6)$$R_{m}=\frac{\Delta P}{\mu \times J_{w}}$$
(7)$$R_{f}=\frac{\Delta P}{\mu \times J_{p3}}-R_{m}$$
(8)$$R_{ir}=\frac{\Delta P}{\mu \times J_{r3}}-R_{m}$$
(9)$$R_{r}=R_{f}-R_{ir}$$
where Δ*P* is the transmembrane pressure (0.1 MPa) and *µ* is the viscosity of pure water. *J_p_*_3_ and *J_r_*_3_ represent the final permeate flux at the end of the third cycle and the recovered pure water flux after the third cycle. 

## 3. Results and Discussion

### 3.1. Chemical Composition of hPAN

The FTIR spectra of the unmodified PAN, hPAN3000, hPAN5000, and hPAN7000 samples are shown in Figure 4. For all the samples tested, the most prominent peaks are found at 2243 cm^−1^ and 1453 cm^−1^, which can be attributed to the stretching of the C≡N bond and the bending of the C-H bond, respectively. These peaks are a result of the original PAN molecules [19]. As the quantity of added hydroxyl ions increased, two peaks emerged gradually at 1578 cm^−1^ and 1541 cm^−1^. The peak at 1578 cm^−1^ can be assigned to C=N stretching that referred to the IR adsorption bands from the laddered PAN structure (a heat-stabilized PAN used as the precursor of carbon fiber) [20]. The peak at 1541 cm^−1^ was assigned to the stretching of N→O coordinate covalent bonds in nitro groups. It was reported that the N→O groups, with the nitrogen atom located in the conjugated C=N bonds, were responsible for the reddish color of the cyclized PAN intermediate [21]. This is consistent with our experimental observation that the color of hPAN dopes shifted from light yellow (hPAN3000) to orange (hPAN5000) and finally to dark red (hPAN7000) as the quantity of added hydroxyl ions increased. It is worth mentioning that no characteristic bands of carboxyl groups at 1410 cm^−1^, the final hydrolysis product of nitrile groups in PAN [22], was found in any of the hPAN samples. This indicates that only the intermediate hydrolysis products of PAN were successfully produced during the in situ pre-hydrolysis treatment.

To further investigate the chemical structure of the hPAN samples, 1H NMR spectra of unmodified PAN and hPAN7000 were analyzed, as presented in Figure 5. Analyzing the NMR spectrum of high molecular weight polymers can be challenging due to weak resonance signal and complex chemical structure. In this study, the average molecular weight of PAN was around 150,000, and the pre-hydrolysis reaction only took place partially, making accurate speculation of the cyclized PAN intermediate even more difficult. However, some evidence of the cyclized PAN structure in hPAN7000 were still found. Two peaks (center of multiplet) at approximately 7.19 ppm and 7.75 ppm were observed in the hPAN7000 1H NMR spectrum and located in the chemical shift range of aromatic structures. Moreover, they were close to the simple heterocyclic nitrogen compound pyridine, with β-H and γ-H having chemical shifts of 7.28 ppm and 7.69 ppm, respectively. 

### 3.2. Membrane Functional Groups

Figure 6a shows the FTIR spectra of the prepared membranes’ surface (blank PAN, M-0, M-1, M-2, and M-3). Compared with the FTIR spectra of the hPAN samples, the membranes that underwent precipitation in NaOH solution showed two new peaks at 1575 cm^−1^ and 1410 cm^−1^, which are related to the asymmetric and symmetric stretching of the carboxyl groups, respectively. The existence of the carboxyl groups in these membranes demonstrates that the hydrolysis of hPAN resumed and progressed further during the phase inversion in NaOH solution. As the quantity of hydroxyl ions added to the hPAN dopes increased, the peak of the carboxyl group in membranes became prominent, indicating that the pre-hydrolysis of PAN could lead to a stronger hydrolysis effect in the following phase inversion in NaOH solution. This may be resulted from the higher reactivity of the cyclized PAN intermediate in hPAN than the nitrile groups in PAN.

In addition, we conducted an analysis of the FTIR spectra of the reconstructed M-0 and M-3 membranes that were first dissolved in DMSO and then precipitated in isopropanol. The results are presented in Figure 6b. It was interesting to find that the peaks of the carboxyl groups were still significant in the M-3 membrane, but not in the M-0 membrane. This demonstrates that the carboxyl groups were successfully embedded within the M-3 membrane structure, showing a more uniform modification. Therefore, this pre-hydrolysis approach could be a novel method for preparing anti-fouling membranes as both the inner and outer surfaces of the membrane could have functional groups.

### 3.3. Membrane Hydrophilicity

The CA values of the prepared membranes are presented in Figure 7. The M-0 membrane shows a significant decrease in its CA value from 45.6° to 19.4°, compared to the blank PAN membrane. This can be explained by the formation of hydrophilic functional groups during the membrane phase inversion in NaOH solution. With the rise of the quantity of hydroxyl ions added to the hPAN dopes, the hydrophilicity of the resulting membranes gradually increased. The M-3 membrane’s CA value was measured as 12.4°, indicating that its surface achieved superhydrophilicity. The results of CA measurement align with the results of FTIR analysis, with both showing that the pre-hydrolysis treatment led to the formation of more hydrophilic functional groups as the intensity of the pre-hydrolysis reaction increased. 

### 3.4. Membrane Morphology

The SEM images shown in Figure 8 present the surface and cross-section morphology of the blank PAN, M-0, and M-3 membranes. As seen from the surface SEM images, all membranes exhibit a uniform surface. The blank PAN membrane surface is porous, and its pores are easily observed. The mean pore size of the blank PAN membrane is estimated to be 0.017 μm. The pores on the surface of the M-0 membrane are still visible, but they have been reduced in size, with the mean pore size estimated to be 0.012 μm. This is due to the pore shrinking effect caused by the phase inversion in NaOH solution [15]. The surface pores of the M-3 membrane cannot be clearly seen, indicating that hCPAN7000 dope resulted in stronger pore shrinkage than the unmodified PAN dope. The blank PAN membrane cross-section shows extensive, finger-like macro-voids, while the cross-section of the M-1 membrane is more uniform with less and straighter macro-voids. The M-3 membrane, on the other hand, has fewer, smaller and more ordered macro-voids. The significant shrinkage in both the surface pores and cross-sectional macro-voids of the M-3 membrane suggest an intense and homogeneous hydrolysis reaction happened while precipitating the hPAN7000 dope in NaOH solution, which may contribute to the resistance of protein adsorption on both the membrane surface and interior pores.

### 3.5. Protein Filtration Performance

Filtration testing of a BSA solution was carried out to assess the protein retention and fouling resistance of the prepared membranes. For this purpose, three membranes—blank PAN, M-0, and M-3—were selected for evaluation. These membranes represent the membrane preparation conditions of no modification, precipitation in NaOH solution only, and pre-hydrolysis treatment combined with membrane phase inversion in NaOH solution, respectively.

As shown in Figure 9 and Table 2, the initial PWF of the blank PAN, M-0, and M-3 membranes were 134.9 L∙m^−2^∙h^−1^, 93.2 L∙m^−2^∙h^−1^ and 80.2 L∙m^−2^∙h^−1^, respectively. The reduction in PWF aligns with the morphological changes, as a higher level of pre-hydrolysis intensity resulted in a greater degree of structural shrinkage.

Although the initial PWF of the blank PAN membrane was the highest, during the first 45-min filtration cycle of the BSA solution, the permeate flux decreased rapidly, reaching a steady and low permeation flux around 57 L∙m^−2^∙h^−1^. The sharp decline was also observed at the beginning of the subsequent filtration cycles. As the filtration cycle progressed, the *RFD* value rose from 57.4% to 61.0%, indicating an increase in BSA fouling due to poor resistance. A decrease was observed in the *FRR* value from around 65.0% to 53.2% after the first and third filtration cycles, respectively. This is a sign of irreversible fouling, resulting in a continual decline in membrane performance.

The M-0 membrane showed improved resistance to flux decline and more efficient flux recovery compared to the blank PAN membrane. The rate of flux decline became more gradual until reaching a steady state. Over the three filtration cycles, the *RFD* value ranged from 27.3% to 32.6% in the filtration cycles. A high flux recovery of 88.4% was observed after the first filtration, although only a limited flux recovery of 83.7% was seen after the third cycle.

The M-3 membrane was found to exhibit the superior flux sustainability and recoverability compared to the other membranes tested. The *RFD* value for the M-3 membrane was in the range of 19.4% to 21.0% only. The rate of flux decline also became slower. In addition, the M-3 membrane demonstrated remarkable flux recovery above 95.2% across the filtration cycles, indicating very low level of irreversible fouling and easy removal of protein deposition through backwashing.

As shown in Figure 10, the BSA rejection rate for the blank PAN membrane increased from 82.1% to 86.9% as the filtration cycles progressed, possibly due to the BSA fouling through pore blocking effect. In comparison, the BSA rejection rates for the M-0 and M-3 membranes were relatively stable over the filtration cycles, ranging from 91.6% to 93.6% and 96.1% to 96.8%, respectively. The more consistent protein retention may be attributed to the small surface pore sizes of these two membranes, as demonstrated in Figure 8, which rendered the protein aggregation mostly on the membrane surface. Among the tested membranes, the M-3 membrane achieved the highest protein retention of over 96% throughout the filtration cycles. The improved protein retention of the M-3 membrane can be attributed to both the size and electrostatic exclusion effects. The pre-hydrolysis reaction combing with precipitation in NaOH solution resulted in a reduction of the M-3 membrane’s surface pore size, while negatively charged carboxylate groups on both the surface and interior pores of the membrane. Due to the deprotonation of carboxyl groups (pKa~3) at a filtration pH of 7.4, the M-3 membrane were able to exclude the negatively charged BSA molecules and prevent their adhesion by electrostatic repulsion. 

The fouling mechanism of the tested membranes was further investigated by the analysis of filtration resistance. The intrinsic resistance (*R_m_*), total fouling resistance (*R_f_*), and reversible (*R_r_*) and irreversible resistance (*R_ir_*) of the tested membranes are shown in Figure 11a, while the partition of the *R_f_* is shown in Figure 11b. The *R_m_* value increased in the sequence of PAN, M-0, M-3, which explains the declining trend of PWF for these membranes in the same order. The *R_r_* values were used to determine the degree of concentration polarization (CP) effect, one of the leading cause for the fouling of porous membrane [23]. The CP effect was significant in this study due to the dead-end filtration mode used, resulting in relatively high *R_r_* values (in comparison to *R_m_* values) of 1.66, 1.11, and 0.96 (×10^−12^ m^−1^) for the PAN, M-0, and M-3 membranes, respectively. Compared with the less varied *R_r_* values, the *R_ir_* values changed a lot. The *R_ir_* value of the M-3 membrane was only 0.23 (×10^−12^ m^−1^), representing only 31% of the M-0 (0.75 (×10^−12^ m^−1^)) and 9% of the PAN membrane (2.51 (×10^−12^ m^−1^)). As the tested membranes had different pore sizes, examining the partition of the *R_f_* could provide some insight of the membrane fouling behavior. Figure 11b reveals a gradual shift in the BSA fouling situation towards more reversible fouling (increasing ratio of *R_r_*/*R_f_*) and less irreversible fouling (decreasing ratio of *R_ir_*/*R_f_*) in the sequence of PAN, M-0, M-3. This trend might be attributed to two possible causes. First, the modified membrane had smaller surface pore sizes, making the reversible CP effect more dominant in filtration resistance. Second, the modified membrane was highly antifouling, leading to a significant reduction in *R_ir_* and thus a higher partition of *R_r_*.

To assess the competitive performance of the M-3 membrane, the filtration results were compared with several published results from commercial and laboratory-made membranes under similar testing conditions. Table 3 presents the pure water permeability, BSA filtration, BSA rejection, flux decline ratio, and flux recovery ratio after membrane cleaning for comparison. Among the listed membranes, the M-3 membrane exhibited high BSA rejection, low *RFD*, and high *RFR*, though the permeability during BSA filtration was restricted to a relatively low initial pure water flux. Therefore, it may be worth exploring a pre-modification strategy that combines the pre-hydrolysis treatment of PAN together with the addition of porogens.

## 4. Conclusions

This study investigated a novel pre-hydrolysis method for the preparation of PAN-based membranes exhibiting low-protein-fouling and high-protein-retention properties. The membranes, which underwent pre-hydrolysis treatment to varying extents, were successfully obtained. Through the chemical characterization of the pre-hydrolyzed PAN dopes, the intermediate hydrolysis product of the cyclized PAN was identified, which was found to promote the subsequent hydrolysis reaction during the membrane phase inversion in the NaOH solution. 

The modification method employed in this study resulted in a more homogeneously hydrolyzed membrane, with hydrophilic and negatively charged carboxyl groups presented not only on the membrane surface, but also within its interior structure. The prepared membrane exhibited a highly hydrophilic surface, with the water contact angle measured to be 12.4° by the captive bubble method. Notably, the membrane demonstrated remarkable antifouling properties in BSA filtration experiment, with a flux decay as low as 21% and a flux recovery as high 95% over three filtration cycles. Additionally, the membrane showed a high BSA rejection rate greater than 96%. These results suggest that the novel pre-hydrolysis method holds great potential for developing PAN-based membranes with enhanced antifouling performance and protein retention. Future research could focus on the utilization of the homogeneously hydrolyzed PAN-based membrane as the base membrane to synthesize diversified functional membranes with possible synergistic effects.

## Figures and Tables

**Figure 1 membranes-13-00310-f001:**
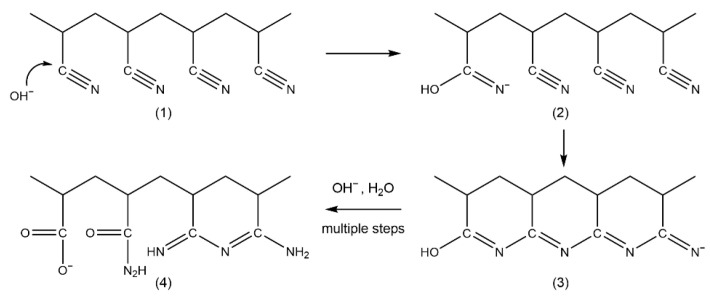
Possible pathway of the cyclized PAN intermediate formation [14]: (1) PAN, (2) nucleophilic substitution of hydroxide ion, (3) cyclized PAN intermediate, (4) functional groups in PAN after hydrolysis.

**Figure 2 membranes-13-00310-f002:**
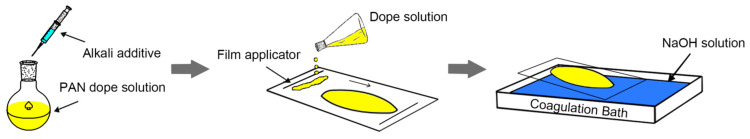
Procedures for preparing the pre-hydrolyzed PAN-based membrane.

**Figure 3 membranes-13-00310-f003:**
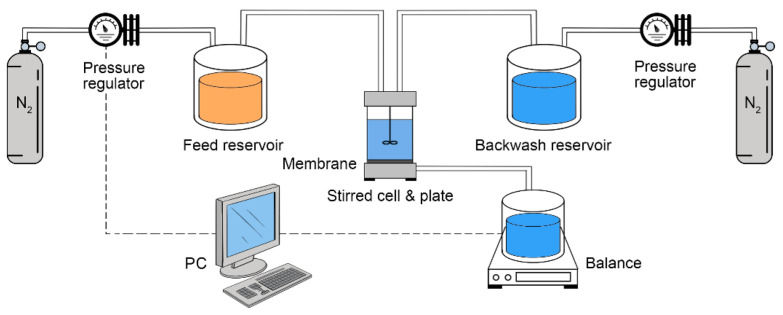
The dead-end filtration system, consisting of two separate paths of forward filtration and backwash.

**Figure 4 membranes-13-00310-f004:**
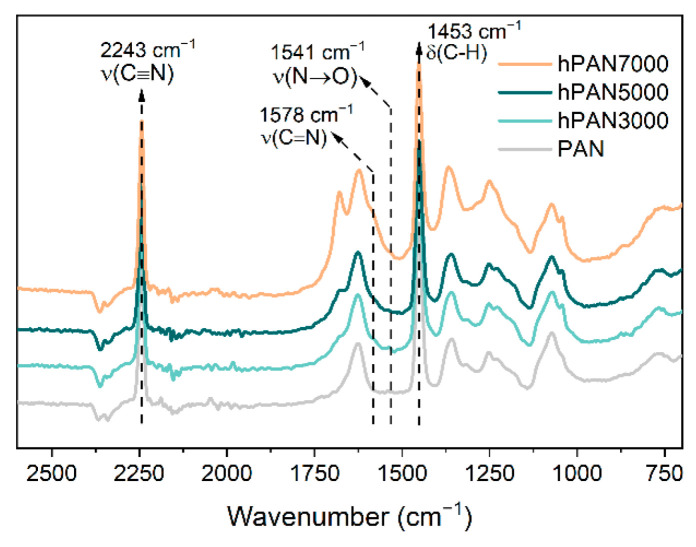
FTIR spectra of the unmodified PAN, hPAN3000, hPAN5000, and hPAN7000.

**Figure 5 membranes-13-00310-f005:**
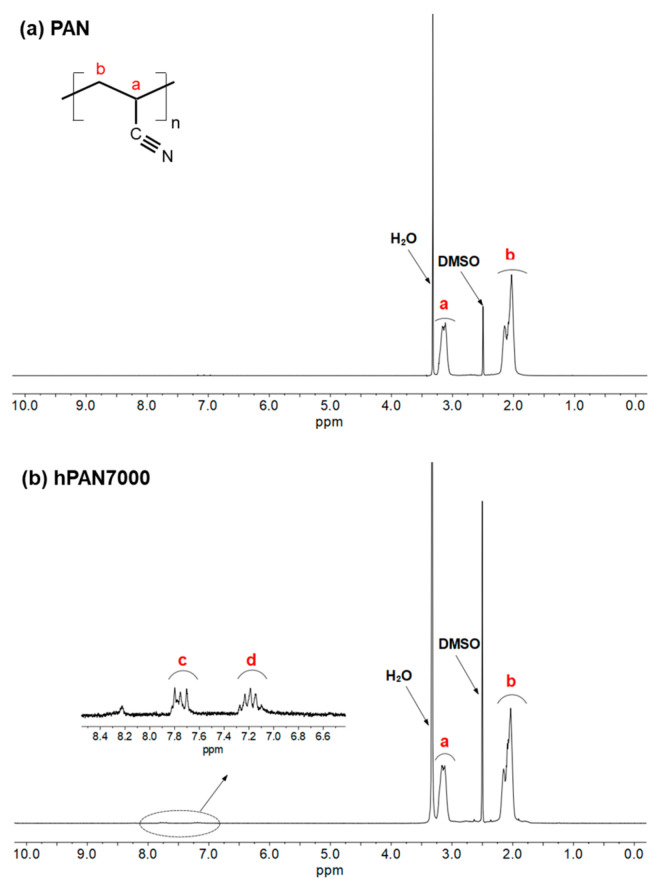
1H NMR spectra of (**a**) unmodified PAN, and (**b**) hPAN7000, in DMSO-d6.

**Figure 6 membranes-13-00310-f006:**
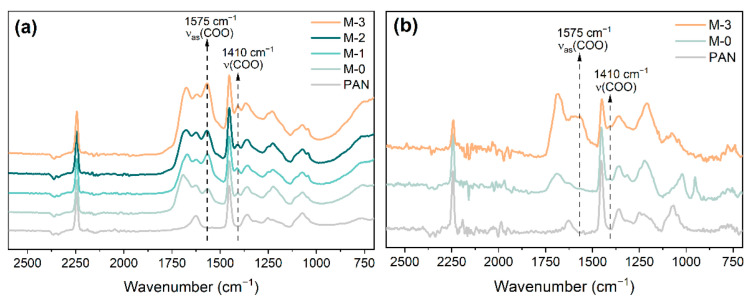
FTIR spectra of the (**a**) surface of the blank PAN, M-0, M-1, M-2, M-3 membranes, and (**b**) the reconstructed blank PAN, M-0 and M-3 membranes.

**Figure 7 membranes-13-00310-f007:**
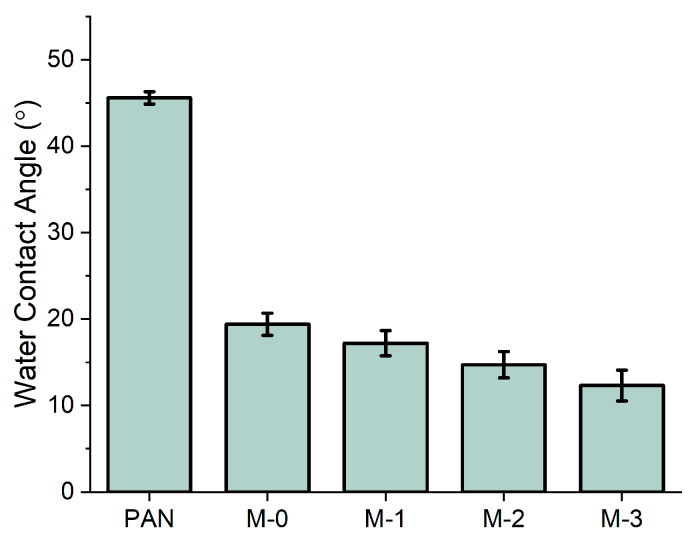
Water contact angle of the blank PAN, M-0, M-1, M-2, and M-3 membranes (measured by captive bubble method).

**Figure 8 membranes-13-00310-f008:**
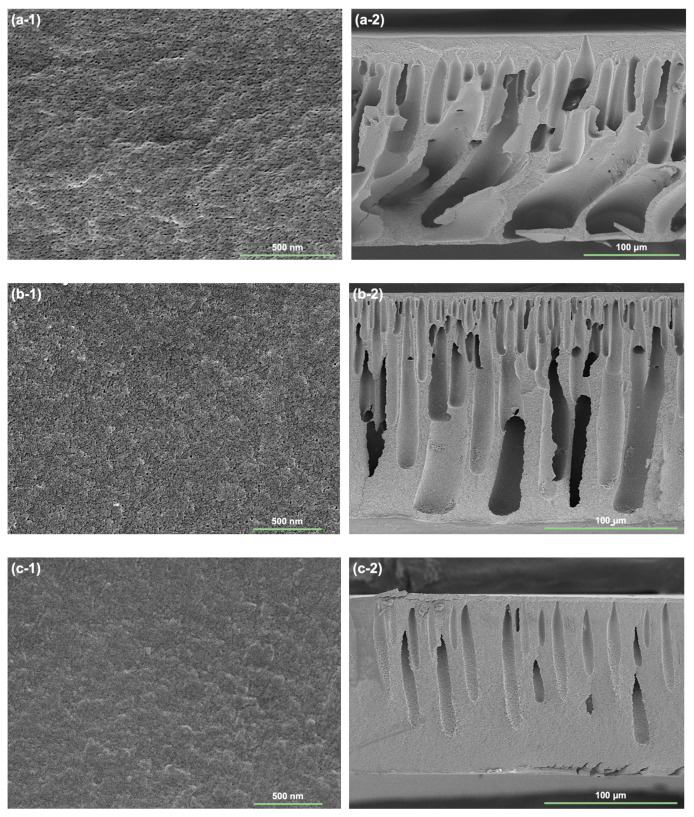
SEM images of the prepared membrane surface (**a-1**–**c-1**) and cross-section (**a-2**–**c-2**): (**a**) blank PAN, (**b**) M-0, and (**c**) M-3.

**Figure 9 membranes-13-00310-f009:**
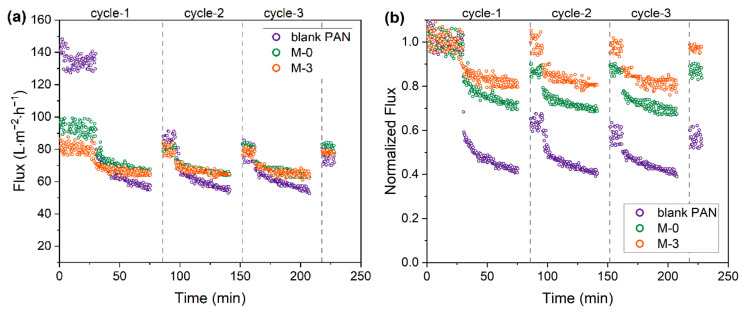
(**a**) Flux and (**b**) normalized flux variation of the blank PAN, M-0, and M-3 membranes.

**Figure 10 membranes-13-00310-f010:**
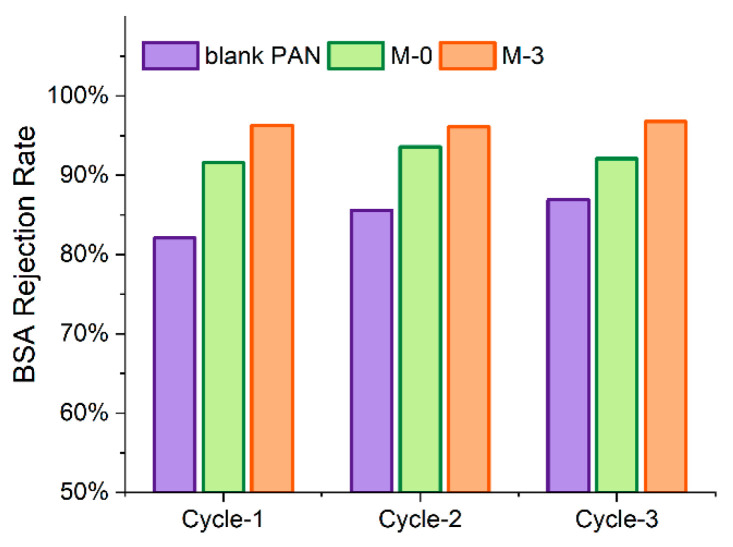
The BSA rejection rate of the blank PAN, M-0 and M-3 membranes across the filtration cycles.

**Figure 11 membranes-13-00310-f011:**
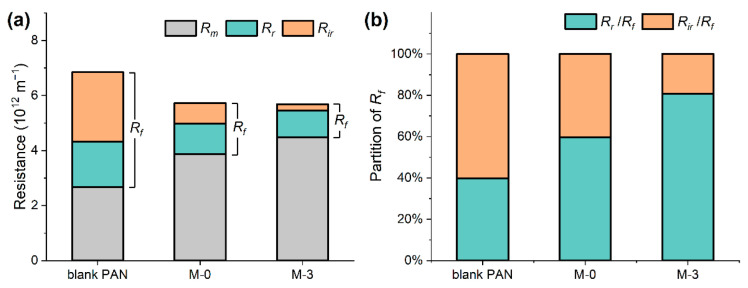
(**a**) Filtration resistance (*R_m_*, *R_r_* and *R_ir_*) of the blank PAN, M-0, and M-3 membranes; (**b**) the partition of the total resistance of fouling (*R_f_*).

**Table 1 membranes-13-00310-t001:** Preparation conditions of membranes.

Code	Dope	Coagulation Bath
blank	Unmodified PAN	DI water
M-0	Unmodified PAN	10% NaOH solution
M-1	hPAN3000	10% NaOH solution
M-2	hPAN5000	10% NaOH solution
M-3	hPAN7000	10% NaOH solution

**Table 2 membranes-13-00310-t002:** Initial pure water flux (*J_w_*), flux at cycle end (*J_p_*), recovered flux (*J_r_*), relative flux decline (*RFD*), relative recovery ratio (*RFR*), and BSA rejection rate (*R*) during 3-cycle BSA filtration test for the blank PAN, M-0, and M-3 membranes.

Membrane	*J_w_* (L∙m^−2^∙h^−1^)	Cycle No.	*J_p_* (L∙m^−2^∙h^−1^)	*J_r_* (L∙m^−2^∙h^−1^)	*RFD*	*RFR*	*R*
Blank PAN	134.9	1	57.5	87.6	57.4%	65.0%	82.1%
		2	54.9	79.1	59.3%	58.6%	85.6%
		3	52.6	69.5	61.0%	53.2%	86.9%
M-0	93.2	1	67.8	82.4	27.3%	88.4%	91.6%
		2	64.1	81.2	31.3%	87.1%	93.6%
		3	62.9	78.0	32.6%	83.7%	92.1%
M-3	80.2	1	63.9	78.0	20.3%	97.2%	96.3%
		2	64.7	76.9	19.4%	95.8%	96.1%
		3	63.4	76.3	21.0%	95.2%	96.8%

**Table 3 membranes-13-00310-t003:** The filtration performance of some commercial and laboratory-made membranes for BSA protein separation.

Membrane	Water Flux (L∙m^−2^∙h^−1^)	BSA Flux (L∙m^−2^∙h^−1^)		BSA Rejection		RFD	RFR	Ref
		High	Low	High	Low			
Amicon PM30	640–1130	160–180	120–140	90–95%	75–80%	79–88%	-	[24]
Modified Alfa Laval membrane	175	150–160	110–120	95–100%	-	47–31%	55–60%	[25]
Osmonics MW	108 (constant flux filtration)			80–92%	20%	-		[26]
PVDF (PEG600)	62.5–100	37.5	25	99%	86.6%	60–62.5%	48–58%	[27]
PVDF (PEG200–20,000)	64–143	-	-	92.6%	80.4%	-	-	[28]
PAN (M-3)	80.2	64.7	63.4	96.8%	96.1%	19.4–21%	95.2–97.2%	This work

## Data Availability

Data are contained within the article.

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
