# Peer review of "Preparation of a Low-Protein-Fouling and High-Protein-Retention Membrane via Novel Pre-Hydrolysis Treatment of Polyacrylonitrile (PAN)"

_membranes, 2023, doi:10.3390/membranes13030310_

Round 1

Reviewer 1 Report

This study examines the preparation, characterization, and application of negatively charged polyacrylonitrile membranes for protein ultrafiltration.  The idea of using negatively-charged membranes to enhance the retention of negatively-charged proteins is well-established, with numerous previous studies of this type.  Thus, the current work is not particularly novel, although this is one of the few studies to examine the premodification of PAN to make these negatively-charged membranes.  The results clearly show that the modified membranes are very hydrophilic with low protein fouling behavior (at least under the conditions of these experiments).  I think the paper is publishable, but only after the authors address the following issues:

1.     The data in Figure 8 clearly show a significant flux decline during the BSA filtration. However, Table 2 only shows an average BSA rejection coefficient.  It would be very useful if the authors could (at a minimum) comment on how the BSA rejection coefficient varies during the filtration as the flux decreases.  This would provide much greater insights into the underlying physical phenomena.

2.     There is no discussion of the effects of concentration polarization on the flux or rejection behavior, even though polarization is likely to be quite high in the dead-end filtration cell.  Some discussion of how polarization might have affected the results is needed.

3.     The authors indicate that “the M-3 membrane achieved impressive protein retention of over 96%.” This is not particularly impressive.  UF processes are typically designed to provide at least 99% and more commonly 99.9% protein retention.  This high level of protein retention is needed since most UF processes also employ a diafiltration, which will cause significant product loss if the protein retention isn’t very high.  I would encourage the authors to more carefully consider this aspect of their work.

4.     There is no comparison of the results obtained in this study with literature data for BSA ultrafiltration.  Is a membrane with 96.5% BSA retention and a pure water flux of 80.2 LMH better than what is currently available from commercial manufacturers?  This is critical to understanding the significance of this work.

5.     It would be helpful if the authors could provide a bit more details on the dead-end filtration cell.  Most dead-end cells cannot be backflushed since the membrane is not supported on the feed-side, but that was clearly not the case in this work.

Reviewer 2 Report

In this manuscript, the authors developed a novel method for preparing hydrophilic and negatively charged PAN membrane for protein retention. The membranes developed here have the potential to be used in various application areas such as food and pharmaceutical industries. The presentation of manuscript is clear.  The study may be of interest to readers of the journal Membranes. I think that although the article has several deficiencies, it can be published after corrections given below.

-       In introduction part, please emphasize the novelty of the study.

-       On page 2, line 65. What “physicochemical characteristics” are the authors mentioning? Briefly describe these properties.

 -       Some texts are not read clearly in some Figures. Please check and revise them. 

-       The contact angle photos shown in Figure 6 are misleading as they do not match the measured values. Please replace them with actual measurement photos or delete them.

Round 2

Reviewer 1 Report

I have reviewed the revised version of the paper and am pleased to recommend the manuscript for publication in Membranes.  The authors have done a nice job addressing my previous concerns.  The additional information on the fouling resistances and the comparison of the performance with that of commercial membranes have definitely strengthened the manuscript.